# Evaluation of Gene Polymorphism and Gingival Crevicular Fluid Levels of Matrix Metalloproteinase-3 in a Group of Turkish Periodontitis Patients

**DOI:** 10.3390/pathogens10101260

**Published:** 2021-09-29

**Authors:** Gökhan Kasnak, Mustafa Yılmaz, Revan Birke Koca Ünsal, Nuray Gürel Polat, Erhan Fıratlı

**Affiliations:** 1Department of Periodontology, Faculty of Dentistry, Istanbul University-Cerrahpaşa, Istanbul 34098, Turkey; 2Department of Periodontology, Faculty of Dentistry, Biruni University, Istanbul 34010, Turkey; myilmaz@biruni.edu.tr; 3Department of Periodontology, Faculty of Dentistry, University of Kyrenia, Kyrenia 99320, Cyprus; revanbirke.koca@kyrenia.edu.tr; 4Virology and Fundamental Immunology Unit, Faculty of Medicine, Istanbul University, Istanbul 34116, Turkey; nuray.polat@istanbul.edu.tr; 5Department of Periodontology, Faculty of Dentistry, Istanbul University, Istanbul 34116, Turkey; erhanfiratli@gmail.com

**Keywords:** periodontal disease, stromelysin-1, single nucleotide polymorphism, matrix metalloproteinase

## Abstract

Introduction: Periodontitis is characterized by the destruction of tooth-supporting tissues. Matrix metalloproteinases (MMPs) play a significant part in the degradation of collagen structure. The gingival crevicular fluid (GCF) levels of MMPs increase with the progression of periodontal inflammation. Polymorphisms can be responsible for high expression of MMPs and can exacerbate the breakdown of collagen structure. This study aims to investigate the effect of MMP-3 -1171 5A/6A polymorphism and the GCF levels of MMP-3 in a group of Turkish periodontitis patients. Materials and Methods: Non-smoking, stage II grade A periodontitis (S II-Gr A) (*n* = 68) and stage II grade B periodontitis (S II-Gr C) (*n* = 64) patients were recruited. Healthy individuals (H) (*n* = 72) without signs of gingivitis or periodontitis served as the control. Venous blood was collected from participants to obtain DNA, and the polymerase chain reaction–restriction fragment length polymorphism (PCR-RFLP) method was used to detect polymorphism. GCF samples were taken to assess MMP-3 levels using an enzyme-linked immunosorbent assay (ELISA). Results: The MMP-3 -1179 5A/6A distribution showed no significant difference between the groups (*p* > 0.05). However, the MMP-3 GCF levels of the S II-Gr C group were higher than those of both the S II-Gr A and H groups (*p* < 0.05), and elevated MMP-3 levels were detected in S II-Gr A compared to H (*p* < 0.05). Conclusion: The MMP-3 GCF levels showed an association with periodontal tissue destruction, although single nucleotide polymorphism was not associated with the S II-Gr C and S II-Gr A groups in the Turkish population.

## 1. Introduction

Periodontitis is a chronic infectious condition that clinically presents a severe, moderate, or mild breakdown of the tooth-supporting hard and soft tissues [1]. Many studies have established that the primary etiology of the disease is the presence of pathogenic bacteria, and the initial onset of the disease can be recognized as physiological inflammation due to the constant presence of dental biofilm [2,3,4]. Unresolved inflammation and the development of the chronic phase of the disease derive from the disruption of the equilibrium between the host response and the dental biofilm, caused by risk factors such as environmental and genetic variations [5]. Single nucleotide polymorphisms (SNPs), which are nucleotide alterations at a specific site in the genome, are the simplest and most prevalent forms of DNA variations in humans. On average, SNPs can be seen in every 1000 nucleotides, and the majority of these differentiations do not affect cellular function. However, some SNPs have been found to play a part in the development of cancer, neurological diseases, and autoimmune disorders. This might offer an explanation for differences in individuals’ immunological responses to pathogenic microorganisms and in their susceptibility to chronic diseases and disorders such as periodontitis [6].

Matrix metalloproteinases (MMPs), which are zinc-dependent endopeptidases, play a crucial role in the degradation of the structural members of the extracellular matrix (ECM), such as collagen, fibronectin, elastin, integrin, laminin, and proteoglycan. MMPs are responsible for the breakdown of the collagen matrix during the degradation of periodontal tissues (alveolar bone and periodontal ligament), and they can affect bone resorption through osteoclast activation and differentiation as well as through direct bone collagen matrix breakdown [7,8] Increased concentrations of MMPs in the gingival crevicular fluid (GCF) have been shown in several studies [5,9,10,11]; moreover, Astolfi et al., and Sorsa et al., have stated in their studies that MMPs can be found in both active and latent forms, either in chronically inflamed gingival tissues or in extracts of gingival tissues [9,10]. It is clear that MMPs play an essential part in the pathogenesis of periodontitis and their levels can be linked to the severity of periodontal tissue damage. A specific genetic polymorphism in the MMPs’ promoter region can modify their expression levels, indicating how this gene might be significant in promoting tissue destruction in the pathogenesis of periodontitis [12,13]. Functional SNPs in the MMP gene can remarkably enhance susceptibility to periodontitis by increasing the secretion of MMPs [14,15].

MMP-3 (stromelysin-1) is a notable proteolytic enzyme among the other MMP types due to its central role in activating the latent MMP types such as MMP-7, MMP-8, MMP-9, and MMP-13, and it has been claimed that MMP-3 has a pivotal function in the initiation of collagen disintegration in periodontitis [5]. The MMP-3 gene is located on chromosome 11q22.2-22.3 and has functional variants resulting from the deletion or insertion of one adenosine allele (5A/6A) at positions -1612 and -1171. Both polymorphisms of MMP-3 have been investigated in patients with periodontitis from various populations, including Caucasian, Asian, and Latin American societies, to analyze the influence of SNPs on MMP-3 enzyme regulation [15,16]. The current results of studies on the MMP-3 promoter gene variation, on the other hand, have revealed poor outcomes in determining its relation to periodontitis. We have hypothesized that in the Turkish population diagnosed with periodontitis, the promoter polymorphism of MMP-3 at the -1179 location is responsible for the high MMP-3 levels in the GCF that lead to exacerbated periodontal tissue destruction. The present work aims to study the 5A/6A single nucleotide polymorphism at the -1171 location in the MMP-3 promoter gene and its effect on the MMP-3 expression in the GCF in a Turkish population diagnosed with periodontitis.

## 2. Results

The demographic profile and clinical data for the study population are presented in Table 1. Compared to periodontally healthy controls, all clinical measures were increased in the S II-Gr C (*p* < 0.01) and the S II-Gr A (*p* < 0.01) groups. Similar to the comparison of PI scores between the S II-Gr A and H groups, the S II-Gr A group’s PI levels were higher than those of the S II-Gr C group (*p* < 0.05). When the PPD, CAL, BOP, and GI scores of the S II-Gr A and S II-Gr C groups were evaluated, no substantial difference was seen between the groups (*p* > 0.05). No statistical difference was observed between groups with regard to age and gender (*p* > 0.05).

The distribution of the genotype frequencies in the MMP-3 -1171 5A/6A of the participants was found in parallel with the Hardy–Weinberg equilibrium (*p* > 0.05). The distribution of the MMP-3 genotypes, the allele frequencies, and the 5A allele carriage rate was found to be higher in the S II-Gr C group than in the H and S II-Gr A groups; however, the difference was not statistically significant (*p* > 0.05). On the other hand, no statistical difference was observed between the S II-Gr A and H groups in the comparison of the MMP-3 genotype distribution (*p* > 0.05). The MMP-3 genotype distribution and allele frequencies are given in Table 2.

We also analyzed the effect of rare and common alleles on the expression of MMP-3 in the S II-Gr C, S II-Gr A, and H groups. The highest MMP-3 secretion was detected in 5A/5A compared with the other alleles of the S II-Gr C group, but it was not statistically significant (*p* > 0.05). Correspondingly, the GCF MMP-3 levels did not show a statistical difference between the rare-allele carriers and the non-carriers of both the S II-Gr A and H groups (*p* > 0.05) (Figure 1).

As presented in Table 3, no significant relationship was identified between the MMP-3 promoter polymorphism and the vulnerability to S II-Gr A and S II-Gr C after adjustment for demographic covariates (OR:1.32 *p* = 0.741 and OR:0.38, *p* = 0.550, respectively).

## 3. Discussion

MMP-3 is responsible for the degradation of proteoglycans, laminin, gelatin, fibronectin, and type IV and type IX collagens. It is also critical for the activation of latent MMPs, such as proMMP-1, proMMP-8, and proMMP-9 [17,18]. Therefore, it is notable that MMP-3 has a key function in the degradation of connective tissue in both health and disease. In periodontitis, its levels in the GCF and the gingival tissues have also been found to be elevated [19]. The effects of genetic variation on periodontal disease were first identified by Kornman et al., and the association between chronic periodontitis and IL-1 gene polymorphism was clearly shown [20]. Later, several studies investigated the role of the promoter gene polymorphisms in the pathogenesis of periodontal diseases in different populations [21,22,23].

This study examined the link between MMP-3 -1171 5A/6A polymorphism and the risk of periodontitis at a slow or rapid rate. We analyzed the MMP-3 levels in the GCF of the Stage II Grade A and Grade C periodontitis patients and of the healthy individuals, to assess whether the polymorphism had any effect. To the best of our knowledge, our study is the first to evaluate the correlation of MMP-3 -1171 5A/6A polymorphism with periodontitis in a sample of the Turkish population. The current investigation found no differences in the genotype and allele distribution of MMP-3 -1171 5A/6A variations between the H, S II-Gr A, and S II-Gr C groups, suggesting that the SNP in the promoter region of the MMP-3 gene might not be an indicator for either S II-Gr A or S II-Gr C among Turkish people. Several studies have investigated the effect of the MMP-3 -1171 5A/6A polymorphism on the predisposition to periodontitis in different populations. While some reports found an association between MMP-3 -1179 5A/6A and periodontitis [9,15,24,25,26], others reported no correlation [16,27,28]. Modifications in the techniques used to test polymorphism, inequalities in sample sizes, variations in the classification/definition of periodontitis, and studies conducted in different races might all be reasons for the contradictory results. The presence of the studies, which suggested that a polymorphism in the -1171 position of the MMP-3 gene may have a role in the expression of MMP-3, directed us to pose the question of whether such an association could be found in the Turkish population. 

In the Turkish population, a limited number of studies report an association between the MMP genotype alterations, such as MMP-1 -1607 1G/2G, MMP-2 753 C/T, MMP-8 799 C/T, and MMP-9 1562 C/T and the susceptibility to periodontitis, and it is highly possible that other types of MMPs might play a role in periodontitis in Turkish individuals, rather than MMP-3 [29,30,31,32,33]. Interestingly, Ustun et al., reported no association between MMP-1 -1607 1G/2G polymorphism and periodontitis in the Turkish population [29], whereas Pirhan et al., suggested that the -1607 2G polymorphic allele could be a reason for susceptibility to severe periodontitis among the Turkish people [30]. The current literature evidently suggests that the influence of the MMP-3 -1171 5A/6A polymorphism on periodontitis susceptibility, in either different races or the same race, is conflicting. One explanation for this disparity could be ethnicity [27], while another could be regulatory systems that modulate the MMPs’ activity [9]. For example, a group of researchers has identified the association between the genetic risk factors of biomarkers (IL-1, IL-1β, IL-1α, IL-4, IL-10) and periodontitis in different populations, whereas others found no connection [34,35].

Periodontitis is a multifactorial disease, and several factors take part in its pathogenesis [36]. It is essential to maintain the balance between destructive enzymes, i.e., MMPs and their regulators, to maintain periodontal health, due to the constant presence of the pathogenic microflora [24,28]. MMPs play a vital role in pathological tissue destruction in several diseases such as periodontitis and in physiological collagen turnover [9,37,38]. The MMP activity on the extracellular matrix might be directed by various processes, including transcription regulation, pro-enzyme activation, and MMP inhibition by its tissue inhibitors (TIMPs) [9,30]. Toyman et al., discovered higher MMP-3 levels in the GCF of periodontitis subjects, demonstrating the role of MMP-3 in periodontal tissue degradation [39], and results obtained from various studies showed the efficacy of non-surgical periodontal therapy in decreasing the MMP enzyme levels in the GCF of patients of various ethnicities with both a slow and rapid rate of periodontitis [30,40,41]. Furthermore, the variation from the 6A allele to the 5A allele at the -1171 location of the MMP-3 gene could lead to elevated expression of the MMP, enhancing periodontal tissue destruction and the susceptibility to the development of periodontitis [25]. In the present study, we additionally investigated the effect of the -1171 5A/6A polymorphism on the MMP-3 GCF levels. We observed that the MMP-3 GCF levels were significantly higher in the S II-Gr C and S II-Gr A groups than in healthy individuals, as found in comparable studies [17,39,40,41,42]. Nevertheless, no difference was revealed between 5A allele carriers and non-carriers, implying that this genetic variant might be unrelated to the MMP-3 GCF changes in Turkish S II-Gr A and S II-Gr C patients. The release and the secretion of proinflammatory cytokines (IL-1β, IL-8, TNF-α, prostaglandins, and proteinases) rely on lipopolysaccharides (LPS) derived from the bacterial membrane. If the presence of pathogenic microflora is not disrupted, inflammation is exacerbated by the involvement of leukocytes, monocytes, and macrophages. Subsequently, periodontal cells such as gingival fibroblasts, oral keratinocytes, osteoblasts, and osteoclasts are activated as a response and proinflammatory cytokines and MMPs are secreted [10]. In accordance with this, an in vitro study performed by Zhou et al., demonstrated the effect of *Porphyromonas gingivalis* on collagen degradation via the induced expression of MMP-3 [42]. Another study conducted in a Turkish sample population also showed increased levels of MMP-3 in the GCF of patients with both aggressive and chronic periodontitis, in the presence of periodontal–pathogenic microorganisms [17]. On the other hand, mechanisms such as COX2 pathways have a reductive function over the MMP-3 transcription [43], which might be a rational explanation for the decrease in the MMP-3 GCF levels after effective periodontal treatment. It is highly possible that one of the previous theories could help to explain the higher MMP-3 levels found in the current investigation. In our view, the MMP-3 transcription might be more dependent on stimuli from pathogenic microorganisms than on the -1171 5A/6A polymorphism for the Turkish individuals.

One limitation of this study is the small sample size, although it is not particularly small compared to other studies in the literature. More research in a broader, more ethnically varied population, is required to determine the significance of the MMP genes in periodontitis risk. The exposure of the carrier gene to a particular environmental factor might provoke the disease, and therefore not including smokers in the study group represents another limitation of the study. Smoking, as a modifier of the host response, is a well-known environmental factor inducing periodontitis [44]; therefore, in future studies, periodontitis patients who are smokers should be included in a study group, to elucidate the association between promoter polymorphism and smoking. Additionally, the effect of single-gene polymorphisms on periodontitis may be questionable due to its multifactorial etiology and the complex structure of the pathogenesis. Nonetheless, the existing data might help plan future studies to evaluate single-/multi-gene polymorphisms in periodontitis patients or to use MMPs as a diagnostic marker via chair-side testing [45].

As a result, MMP-3 -1171 5A/6A polymorphism showed no effect on the S II-Gr A and S II-Gr C groups in a sample of the Turkish population. In relation to the clinical status, the MMP-3 GCF levels were higher in the periodontally diseased groups than in the control group. Our findings suggest that MMP-3 is a biomarker associated with periodontitis, and its expression might be dependent on several other immunological mechanisms as well as the genetic polymorphisms, for Turkish individuals. Further research could help us understand not only the pathophysiology of periodontitis but also that of other multifactorial diseases. Early identification methods for high-risk individuals can thus be developed.

## 4. Materials and Methods

### 4.1. Study Population and Clinical Assessment

The study included patients with periodontitis (69 males and 63 females aged 26–37 years) referred to the University of Istanbul Faculty of Dentistry Department of Periodontology, with 72 periodontally healthy (H) individuals (37 men and 35 females ranging in age from 28 to 35 years) serving as controls. All the participants were physically and mentally healthy, and none of them had received previous periodontal treatment. Active caries lesions, oral mucosal problems, active orthodontic therapy, pregnancy, lactation, smoking, and the use of prescribed or non-prescribed medications or antibiotics for at least six months prior to recruitment were all exclusion factors. The University of Istanbul’s Faculty of Dentistry’s Ethics Committee approved the study protocol, which followed the principles of the Declaration of Helsinki (2017/41).

A single calibrated examiner (G.K.) performed the periodontal examinations using a periodontal probe (UNC-15, Hu-Friedy, Chicago, IL, USA) to measure the plaque index (PI),18 gingival indexes (GI),19 clinical attachment levels (CAL), bleeding on probing (BOP),20 and probing pocket depth (PPD) at four sites on all teeth, as well as alveolar bone loss measurements via orthopantomography. All the individuals’ orthopantomographs were obtained using an extra-oral diagnostic system (KODAK 9000 3D, Carestream Dental LLC, Atlanta, GA, USA and Dental Imaging Software CS 3D, Carestream Dental LLC, Atlanta, GA, USA) and analyzed using computer software built exclusively for the storage and analysis of digital data received from the X-ray machine. The distance between the cementoenamel junction and the most coronal area of the bone defect was measured on both the mesial and distal sites for the control and periodontitis groups, and the ratio of this to the length of the root was calculated to evaluate the bone loss.

Periodontitis patients were grouped according to the 2017 World Workshop on the Classification of Periodontal and Peri-Implant Diseases and Conditions [46]. Sixty-eight periodontitis patients (35 males and 33 females aged 29–37), who had moderate periodontal destruction in parallel with the presence of microbial dental biofilm, were classified as Stage II Grade A (S II-Gr A), while 64 periodontitis patients (34 males and 30 females aged 26–34) with clinical signs of rapid periodontal destruction accompanied by molar/incisor pattern periodontitis were classified as Stage II Grade C (S II-Gr C).The participants in this study were all unrelated and came from various parts of Turkey.

### 4.2. Genetic Analysis

Nine millilitres of venous blood was drawn into EDTA coated tubes (Greiner Bio-One™ VACUETTE™ K3EDTA Blood Collection Tubes, Fisher Scientific, Hampton, NH, USA) from each individual by the same experienced medical staff member, and the genomic DNA was extracted from peripheral blood leukocytes using the standard proteinase K digestion method. The polymorphism of MMP-3 -1171 5A/6A was identified using the polymerase chain reaction–restriction fragment length polymorphism (PCR-RFLP) technique, as reported previously by Astolfi et al., Briefly, PCR was performed in a 25 mL volume containing 400 ng of genomic DNA, 10 mM Tris-HCl (pH 8.3), 50 mM KCl, and 1.5 mM MgCl, 2.1 mM primers, 200 mM dATP, dCTP, dGTP, and dTTP, and 2.5 U of Taq DNA polymerase (Qiagen, Hilden, Germany). In order to amplify the MMP-3 -1171 5A/6A (rs35068180) the primers F:5′-GAAGGAATTAGAGCTGCCACA-3′ and R:5′-AAGGGATTTCTCTGTGGCAA-3′ were used. To perform the thermal cycling, the solution was incubated for 3 min at 95 °C, followed by 35 cycles of 1 min at 95 °C, 1 min at 41 °C (6A allele), 1 min at 44 °C (5A allele) and 1 min at 72 °C, with a final 7 min extension at 72 °C. The amplified DNA fragments were digested by Tth111I (Thermo Fisher, Waltham, MA, USA) and on digestion the 5A allele of MMP3 formed 97 and 33 bp products, whereas the 6A allele was not digested and retained its original length of 130 bp. The gel electrophoresis method was performed on a 12% polyacrylamide gel for the visualization of the base pairs of the 5A and 6A alleles. 

### 4.3. Gingival Crevicular Fluid Collection

A single calibrated clinician (G.K.) collected the GCF samples from the distal or mesial sites with each participant’s maximum gingival pocket depth. In the control group, however, the GCF samples were collected from a random crevice of each periodontally healthy individual. The GCF samples were collected using periopaper strips (Oraflow Inc. Hewlett, NY, USA). The strips were placed in separate Eppendorf tubes and labeled and weighed before and after GCF collection. The GCF samples’ net weights were calculated by subtracting the initial weights from those obtained. Samples were kept at −80 °C until the day the ELISA tests were performed. 

### 4.4. ELISA Analysis

The concentrations of MMP-3 were determined using a commercially available ELISA kit (KAC1541-HU MMP-3) according to the manufacturer’s instructions (BioSource, Invitrogen, Waltham, MA, USA). The process began by thawing the frozen Eppendorf tubes by keeping them at a constant room temperature for 20 min. Each GCF sample was treated with 300 microliters (μL) of phosphate-buffered saline (pH 7.0) containing 0.05% of bovine albumin and kept at +4 °C for 24 h. In the next phase, each tube received 100 μL of a 1% BSA-PBS Tween buffer solution (pH 7.4) and was stored at room temperature for one hour before being shaken horizontally at +4 °C for 20 h, and then the periopaper strips were discarded and 100 μL of solution collected from every tube to analyze the MMP-3 levels. GCF MMP-1 values were observed using a commercial ELISA kit according to the manufacturer’s instructions. The two-step sandwich ELISA method was used, where 100 μL standards and samples were placed into microwells coated with the MMP-3 monoclonal antibodies. Multiwell plates were left for four hours for incubation. After the incubation period, the plates were washed, 200 μL of secondary MMP-3 monoclonal antibodies were added and they were incubated for 16 h. Following that, the plates were re-washed, and 200 μL of the chromogenic substrate was added. H_2_SO_4_ was added 15 min later to stop the reaction, and the enzymatic activity was measured at 405 nm. The ELISA results for the MMP were obtained as concentrations in picograms/mL (pg/mL).

In order to calculate the amount of MMP-3 in the GCF samples, the recorded weights of the periopaper strips with absorbed GCF needed to be converted to units of volume. For this reason, a calculation method was carried out as previously described by Yılmaz et al., [17]. Concisely, in the first place, 0.05% of bovine albumin was dissolved in PBS and 100 μL of this solution was added to each of the paper strips, which were then placed in their respective Eppendorf tubes. They were all weighed before and after the procedure. This experiment was carried out ten times for each tube. The unit volume corresponding to the unit weight was calculated by taking the average of these ten measurements. It was determined that 1 g was equivalent to 990 μL. The raw data values were then multiplied by the coefficients to produce accurate data.

### 4.5. Data Analysis

Chi-squared tests were used to examine the frequencies of the MMP-3 -1171 5A/6A SNPs in the H, S II-Gr A, and S II-Gr C groups, as well as deviations in genotype distribution from those predicted by the Hardy–Weinberg equilibrium. 

The differences in the clinical parameters between the study groups were analyzed using the Mann–Whitney U test with the Bonferroni correction, while the relationship between the genotype and the clinical status was examined via multiple logistic regression analysis adjusted for age and gender.

The MMP-3 expression level difference between individuals with or without promoter polymorphism in the GCF of the study groups, was analyzed via ANOVA. 

All the statistical calculations were performed using statistical software (SPSS 21.0, SPSS Inc., IBM, Chicago, IL, USA), and the significance was defined as a *p*-value ≤ 0.05.

## Figures and Tables

**Figure 1 pathogens-10-01260-f001:**
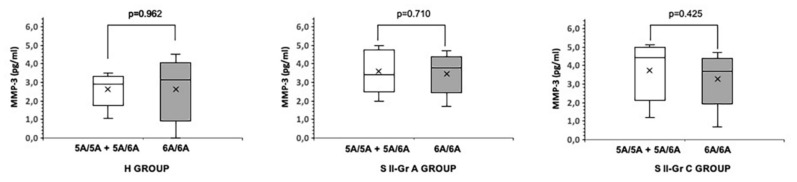
The GCF MMP-3 levels were given according to the rare allele carriage for the healthy individuals (H group), the patients with Stage II Grade A periodontitis (S II-Gr A group), and those with Stage II grade C periodontitis (S II-Gr C group). No statistical significance was detected between the 5A/5A + 5A/6A and 6A/6A subgroups of each group (*p* > 0.05).

**Table 1 pathogens-10-01260-t001:** Characteristics of the study population. Data were given as means ± standard deviations and *p* < 0.05 was considered as significant.

		Study Groups			*p* Value		
		H	S II-Gr A	S II-Gr C	H vs. S II-Gr A	H vs. S II-Gr C	S II-Gr A vs. S II-Gr C
Clinical Measures, Age, and Gender	Age (years) (Mean (Min–Max))	31.8 (28–35)	33.7 (29–37)	30.7 (26–34)	*p* = 0.598	*p* = 0.768	*p* = 0.632
	Gender (Female % (n))	48.6% (35)	48.5% (33)	46.8% (30)	*p* = 0.788	*p* = 0.539	*p* = 0.572
	PPD (mm)	2.6 ± 1.16	4.14 ± 1.02	4.78 ± 1.29	*p* < 0.01	*p* < 0.01	*p* = 0.198
	CAL (mm)	2.81 ± 0.85	4.78 ± 1.84	4.87 ± 1.71	*p* < 0.01	*p* < 0.01	*p* = 0.136
	PI	0.48 ± 0.60	2.67 ± 0.39	1.46 ± 0.44	*p* < 0.01	*p* < 0.01	*p* = 0.029
	GI	0.44 ± 0.52	2.24 ± 0.19	2.19 ± 0.18	*p* < 0.01	*p* < 0.01	*p* = 0.374

PPD: Probing pocket depth, in millimeters; CAL: Clinical attachment level, in millimeters; PI: Plaque index; GI: Gingival index.

**Table 2 pathogens-10-01260-t002:** Distribution of MMP-3 genotypes and alleles. No statistical difference was observed in the comparisons between the groups (*p* > 0.05).

		Rate % (*n*)			$p\;\mathrm{Value}\;(\chi^{2}\mathrm{Test})$		
		H	S II-Gr A	S II-Gr C	H vs. S II-Gr A	H vs. S II-Gr C	S II-Gr A vs. S II-Gr C
MMP-3 Genotype (rs35068180)	5A/5A	1.38% (1)	1.47% (1)	3.12% (2)	0.973	0.054	0.063
	5A/6A	26.38% (19)	26.47% (18)	34.37% (22)			
	6A/6A	72.22% (52)	72.05% (49)	62.50% (40)			
Allele Frequency	5A	14.58% (21)	14.70% (20)	19.40% (26)	0.918	0.084	0.098
	6A	85.41% (123)	85.29% (116)	80.59% (108)			
Carriage Rate	5A+	27.77% (20)	27.94% (19)	37.50% (24)	0.946	0.052	0.053
	5A−	72.22% (52)	72.05% (49)	62.50% (40)			

**Table 3 pathogens-10-01260-t003:** Logistic regression analysis to test susceptibility to periodontitis. Logistic regression analysis showed no difference in the association between MMP-3 genotypes and susceptibility to periodontal disease (stage II grade A and grade C periodontitis), after adjusting for covariates (*p* > 0.05).

	Stage II Grade A				Stage II Grade C			
	Age		Gender	Allele Carriage (5A/5A and 5A/6A)	Age		Gender	Allele Carriage (5A/5A and 5A/6A)
	<30 Years	>30 Years			<30 Years	>30 Years		
OR	2.39	5.58	8.80	1.32	4.40	8.55	9.46	0.38
95% CI	0.13–42.30	0.31–98.76	0.49–155.77	0.07–23.36	0.24–77.87	0.48–151.32	0.53–167.43	0.02–6.72
*p* Value	0.142	0.460	0.838	0.741	0.689	0.128	0.758	0.550

## Data Availability

The data of this study are available on request.

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
