# Peer review of "Evaluation of Gene Polymorphism and Gingival Crevicular Fluid Levels of Matrix Metalloproteinase-3 in a Group of Turkish Periodontitis Patients"

_pathogens, 2021, doi:10.3390/pathogens10101260_

Round 1

Reviewer 1 Report

Introduction, the hypothesis section line 77-78, "....studied in a Turkish population with Chronic periodontitis " should be periodontitis.

It is not clear why authors did not study the second polymorphism (both polymorphisms) which could have provided better explanation to the research question they are posing?

Major problem with the study is authors are not using the current classification system for periodontal diseases and conditions, with staging and grading diagnostic criteria. This needs to be addressed.

The discussion is long and not focused, it reads as a summary of polymorphism studies done on MMP-3 in several populations but it does not provide a sound explanation to the observed differences in this study. There are inherent challenges in methodology, to explain that one gene polymorphism would be responsible for protein production and eventually affect clinical presentation of disease could be difficult.

Table 1, please indicate age in mean average (years) and also in range (years).

Table 3, what is the rationale that age is presented/divided into two, <30 years  and >30 years? Why 30 years of age is important/critical in periodontal disease? I encourage authors to follow the new classification guidelines.

Author Response

The authors appreciate the reviewer for the valuable comments to improve the manuscript. The edited parts have been highlighted, and the revised manuscript has been uploaded. The responses of the authors to the helpful comments of the reviewer are given below.

Kind Regards.

  1. Introduction, the hypothesis section line 77-78, "....studied in a Turkish population with Chronic periodontitis " should be periodontitis.

The word “chronic” was mistakenly written and it is deleted.

  1. It is not clear why authors did not study the second polymorphism (both polymorphisms) which could have provided better explanation to the research question they are posing?

Several studies and meta-analyses have reported -1179 5A/6A rather than the second polymorphism in MMP-3; therefore, the authors focused on the association between the -1179 5A/6A polymorphism in MMP-3 and periodontitis (Weng H, Yan Y, Jin YH, Meng XY, Mo YY, Zeng XT. Matrix metalloproteinase gene polymorphisms and periodontitis susceptibility: a meta-analysis involving 6,162 individuals. Sci Rep. 2016 Apr 20;6:24812. doi: 10.1038/srep24812. PMID: 27095260; PMCID: PMC4837403; Li W, Zhu Y, Singh P, Ajmera DH, Song J, Ji P. Association of Common Variants in MMPs with Periodontitis Risk. Dis Markers. 2016;2016:1545974. doi: 10.1155/2016/1545974. Epub 2016 Apr 19. PMID: 27194818; PMCID: PMC4853955.;da Silva MK, de Carvalho ACG, Alves EHP, da Silva FRP, Pessoa LDS, Vasconcelos DFP. Genetic Factors and the Risk of Periodontitis Development: Findings from a Systematic Review Composed of 13 Studies of Meta-Analysis with 71,531 Participants. Int J Dent. 2017;2017:1914073. doi: 10.1155/2017/1914073. Epub 2017 Apr 26. PMID: 28529526; PMCID: PMC5424192.). On the other hand, the authors agree with the reviewer about the importance of the second polymorphism, and this issue was addressed in the text. 

  1. Major problem with the study is authors are not using the current classification system for periodontal diseases and conditions, with staging and grading diagnostic criteria. This needs to be addressed.

The authors agree with the reviewer. We re-evaluated the clinical records of each patient and re-classified them according to the 2017 classification.  According to the new classification, the chronic periodontitis patients were classified as Stage II Grade A, while the aggressive periodontitis patients were classified as Stage II Grade C (Papapanou PN, Sanz M, Buduneli N, Dietrich T, Feres M, Fine DH, Flemmig TF, Garcia R, Giannobile WV, Graziani F, Greenwell H, Herrera D, Kao RT, Kebschull M, Kinane DF, Kirkwood KL, Kocher T, Kornman KS, Kumar PS, Loos BG, Machtei E, Meng H, Mombelli A, Needleman I, Offenbacher S, Seymour GJ, Teles R, Tonetti MS. Periodontitis: Consensus report of workgroup 2 of the 2017 World Workshop on the Classification of Periodontal and Peri-Implant Diseases and Conditions. J Clin Periodontol. 2018 Jun;45 Suppl 20:S162-S170. doi: 10.1111/jcpe.12946. PMID: 29926490.).

  1. The discussion is long and not focused, it reads as a summary of polymorphism studies done on MMP-3 in several populations but it does not provide a sound explanation to the observed differences in this study. There are inherent challenges in methodology, to explain that one gene polymorphism would be responsible for protein production and eventually affect clinical presentation of disease could be difficult.

The aim of the authors was to highlight the result’s discrepancy in the literature and to compare the studies with the current study because no information is available in the present literature, up to our best of knowledge, related to the MMP-3 promoter polymorphism and periodontitis in Turkish population. The authors also, agree with the reviewer, and the discussion part of the manuscript was shortened and revised. The questionable effect of single-gene polymorphism on periodontal disease was addressed and given as a limitation of the study.

  1. Table 1, please indicate age in mean average (years) and also in range (years).

Table 1 is revised according to the comments of the reviewer. The mean of the age and its range were given.

  1. Table 3, what is the rationale that age is presented/divided into two, <30 years and >30 years? Why 30 years of age is important/critical in periodontal disease? I encourage authors to follow the new classification guidelines.

In the manuscript's first draft, the study population was grouped according to the 1999 periodontal disease classification, which the reviewer also stated. Therefore, in Table 3, 30 years of age was accepted as a diagnostic criteria for periodontal disease according to a consensus report published in 1999 by Lang et al. (Lang, N. et al., Consensus Report: Aggressive Periodontitis. Annals of Periodontology 1999;4(1),53–53). The authors totally agree with the reviewer, and we are thankful for the encouragement of the reviewer.  We re-grouped the study population according to the 2018 classification, as we remarked in response to the third comment (Chronic periodontitis patients > Stage II Grade A; Aggressive periodontitis patients > Stage II Grade C), and revised Table 3.

Reviewer 2 Report

This study evaluated the correlation between gene polymorphism and GCF levels of MMP3 in a group of Turkish periodontitis patients. Methodology and results were well presented and clear. Results were discussed with relevant evidence from literature. Hypothesis was missing in the introduction which I believe is utmost important and it should be well stated.  

Minor spell check and grammatically errors evident in the manuscript.

Another limitation is small sample size which I think authors discussed this.

Author Response

The authors are thankful to the reviewer for the valuable comments to improve the manuscript. The edited parts have been highlighted, and the revised manuscript has been uploaded. The responses of the authors to the helpful comments of the reviewer are given below.

Kind regards.

  1. This study evaluated the correlation between gene polymorphism and GCF levels of MMP3 in a group of Turkish periodontitis patients. Methodology and results were well presented and clear. Results were discussed with relevant evidence from literature. Hypothesis was missing in the introduction which I believe is utmost important and it should be well stated.  

At the end of the introduction section, the hypothesis of the study is given as "We have hypothesized that in the Turkish population diagnosed with periodontitis, the promoter polymorphism of MMP-3 at the -1179 location is responsible for the high MMP-3 levels in the GCF that leads to exacerbated periodontal tissue destruction" (Page 2; lines 75-78).

2. Minor spell check and grammatically errors evident in the manuscript.

The manuscript is checked to eliminate spelling and grammatical errors.  

3. Another limitation is small sample size which I think authors discussed this.

As the reviewer mentioned, we addressed the small sample size as a study limitation in the discussion section. In addition to this, we remarked on the importance of smoking and the questionable role of single-gene polymorphisms on the severity of periodontitis.
